# Distinguishing the impacts of ozone and ozone depleting substances on the recent increase in Antarctic surface mass balance

Rei Chemke[1], Michael Previdi[2], Mark R. England[3,4], and Lorenzo M. Polvani[1,2]

[1]Department of Earth and Planetary Sciences, Weizmann Institute of Science, Rehovot, Israel
[2]Lamont-Doherty Earth Observatory, Columbia University, Palisades, NY 10964, USA
[3]Department of Climate, Atmospheric Science, and Physical Oceanography, Scripps Institution of Oceanography, UCSD, La Jolla, CA, USA
[4]Department of Physics and Physical Oceanography, University of North Carolina Wilmington, NC, USA

**Correspondence:** R. Chemke (rc3101@columbia.edu)

**Abstract.** The Antarctic surface mass balance (SMB) has global climatic impacts through its effects on global sea-level rise. The forced increase in Antarctic SMB over the second half of the 20th century was argued to stem from multiple forcing agents, including ozone and ozone-depleting substances. Here we use ensembles of fixed-forcing model simulations to quantify and contrast the contributions of stratospheric ozone, tropospheric ozone and ozone-depleting substances (ODS) to increases in the Antarctic SMB. We show that ODS and stratospheric ozone make comparable contributions, and together account for 44% of the increase in the annual mean Antarctic SMB over the second half of the 20th century. In contrast, tropospheric ozone has an insignificant impact on the SMB increase. A large portion of the annual mean SMB increase occurs during Austral summer, when stratospheric ozone is found to account for 63% of the increase. Furthermore, we demonstrate that stratospheric ozone increases the SMB by enhancing the meridional mean and eddy flows towards the continent, thus converging more water vapor over the Antarctic.

## 1 Introduction

Being the largest freshwater reservoir on Earth, the Antarctic ice sheet is potentially the largest contributor to future global sea level rise (IPCC, 2013; Fretwell et al., 2013). By 2100, the projected loss of Antarctic land-ice due to dynamical processes (i.e., the flow of the ice sheet) will increase sea level by up to 185 mm (IPCC, 2013; Golledge, 2020). In contrast, the projected increase in Antarctic surface mass balance (SMB, i.e., the larger increase in precipitation vs. evaporation/sublimation; e.g., see Agosta et al., 2019) will reduce this sea level rise by 20-80-mm (Krinner et al., 2007; Uotila et al., 2007; Ligtenberg et al., 2013; Frieler et al., 2015; Previdi and Polvani, 2016; Palerme et al., 2017). Over recent decades, however, only the dynamical mass loss due to the acceleration of outlet glaciers has been documented (Rignot et al., 2004; Shepherd et al., 2012; Velicogna et al., 2014; Wouters et al., 2015; Rignot et al., 2019); the Antarctic SMB has exhibited insignificant trends (Monaghan et al., 2006a, b; Lenaerts et al., 2012), and thus has yet to mitigate sea-level rise. Although climate models do simulate an increase in Antarctic SMB over recent decades in response to the external forcings (Krinner et al., 2007; Monaghan et al., 2008; Palerme

et al., 2017; Previdi and Polvani, 2017), such an increase has been obscured by the large climate variability (Previdi and Polvani, 2016), resulting in insignificant observed Antarctic SMB trends (Frezzotti et al., 2013).

Similar to most late 20th century forced changes in the Southern Hemisphere (Polvani et al., 2011b), the modeled increase in Antarctic SMB, and incursion of dust particles into the Antarctic continent (Cataldo et al., 2013), has also been attributed to anthropogenic emissions. In particular, two recent studies have argued for the importance of increases in the emissions of ozone-depleting substances (ODS) (Previdi and Polvani, 2017) and in stratospheric ozone depletion (Lenaerts et al., 2018) for Antarctic SMB. However, neither study cleanly isolated the effects of these forcing agents: Previdi and Polvani (2017) did not separate the effects of ODS from stratospheric ozone, while Lenaerts et al. (2018) did not separate the effects of stratospheric from tropospheric ozone. This, together with the fact that different seasons were analyzed in these two studies, has prevented a clear attribution of the forced SMB increase. In addition, a quantitative analysis of the mechanism underlying the forced increase in SMB has not been conducted to date.

The aim of this paper is thus to elucidate which forcing agents related to ozone have mostly contributed to the forced increase in Antarctic SMB over the second half of the 20th century, and to quantitatively analyze the underlying mechanisms. This is done using ensembles of fixed-forcing simulations which allow us to disentangle the impacts of stratospheric ozone depletion, tropospheric ozone changes, and ODS emissions on Antarctic SMB trends.

## 2   Methods

We analyze four ensembles of model simulations using the Community Earth System Model (CESM1) (Hurrell et al., 2013): each ensemble is forced with slightly different agents. We use this CMIP5-class model as previous work showed that the CESM captures the spatial patterns of climatological mean Antarctic SMB, and its variability, from ice-cores and reanalysis (Lenaerts et al., 2018). Furthermore, the CESM well captures the climate response in the Southern Hemisphere to ozone-depletion (England et al., 2016; Landrum et al., 2017). This provide us confidence in using the CESM to investigate SMB changes under forced ozone changes.

The first ensemble consists of 10 members, which were randomly picked from the 40-member large-ensemble (LENS) described in Kay et al. (2015). Each member is forced, from 1920 to 2005, with all known natural and anthropogenic forcings, following the Historical specifications of the Climate Modeling Intercomparison Project, Phase 5 (Taylor et al., 2012). While all simulations are subjected to the same forcing, they differ in their initialization: each member is initialized at 1920 with a slightly different air temperature ($\mathcal{O}10^{-14}\,\mathrm{K}$, across all grid points), thus allowing us to investigate the climate's transient response to the external forcing in the presence of internal climate variability.

The other three ensembles are identical to the LENS, but without time evolution of specific forcing agents: a 10-member ensemble with fixed ODS (Fix-ODS, Polvani et al., 2020), a 10-member ensemble with fixed ODS and stratospheric ozone (Fix-ODS&O3$_{\mathrm{strat}}$, Polvani et al., 2020), and an 8-member ensemble with fixed stratospheric and tropospheric ozone (Fix-O3, England et al., 2016; Landrum et al., 2017; Lenaerts et al., 2018). These forcing agents are fixed at 1955 values (pre-ozone hole). Each simulation in the fixed forcing ensembles is initialized from the corresponding simulation of the LENS at 1950.

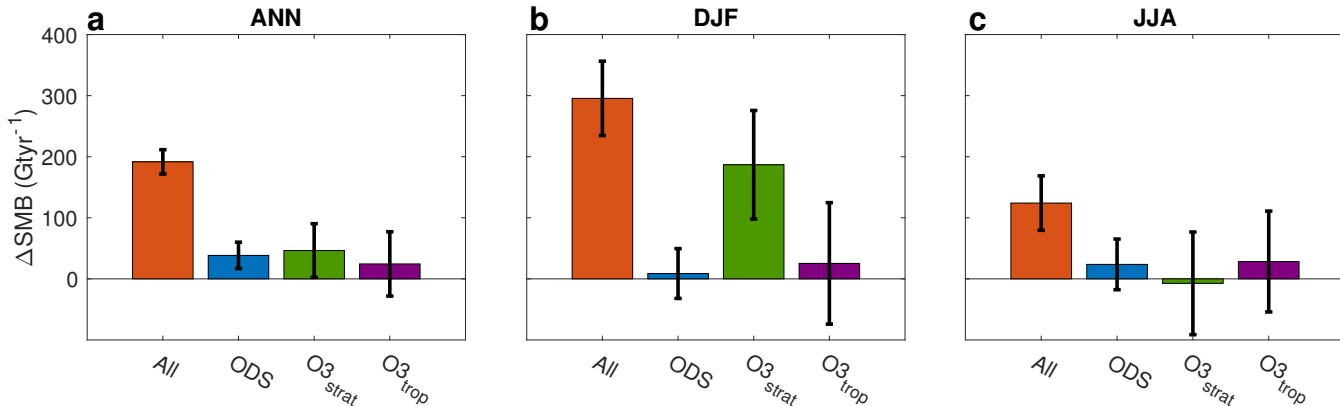

**Figure 1.** Antarctic SMB response to second half of the 20th century forcing (ΔSMB) in (a) annual mean, (b) DJF and (c) JJA. Red bars show the response in the full forcing ensemble mean (All). The blue, green and purple bars show the contributions of ODS, stratospheric ozone ($O3_{strat}$), and tropospheric ozone ($O3_{trop}$) to the SMB response, respectively. Error bars represent the 95% confidence interval.

Thus, the ensemble mean difference between LENS and Fix-ODS, averaged over 1990-2005, allows one to isolate and quantify the effects of ODS over the second half of the 20th century. Similarly, the ensemble mean difference between Fix-ODS and Fix-ODS&$O3_{strat}$ isolates the effects of stratospheric ozone ($O3_{strat}$). Finally, subtracting the effects of stratospheric ozone from the ensemble mean difference between LENS and Fix-O3 isolates the effects of tropospheric ozone ($O3_{trop}$). This procedure assumes that the forcings are linearly additive and, it should be clear, the model runs analyzed here were not originally designed

for this study; rather, we are here simply exploiting these "ensembles of opportunity" which were designed for earlier studies (Polvani et al., 2020; England et al., 2016) to study the effect of three distinct forcing agents on the Antarctic SMB. It should be noted that our approach assumes that the SMB responses to different forcings are additive, and thus that the effects of non-linear interactions between forcings can be ignored (Levermann et al., 2020).

Throughout the manuscript, $\Delta$ represents the response of the climate system to the forcings, which is defined as the difference

between the 1990-2005 period in each ensemble member and the 1940-1955 period in the corresponding member of the LENS. Thus, the response in LENS accounts for the effects of all Historical forcing agents (hereafter referred to as All). We choose the 1990-2005 and 1940-1955 periods in order to examine the SMB response to historical forcing during the second half of the 20th century, when the entire response to ODS/ozone forcing occurs.

## 3   The role of ozone vs. ODS in increasing the annual mean Antarctic SMB

We start by quantifying the contribution of each forcing agent to the Antarctic annual mean $\Delta$SMB (Fig. 1a). Recall that while stratospheric ozone impacts the troposphere, primarily, by shifting the eddy-driven jet towards the South pole in summer, the impact of ODS and tropospheric ozone is primarily a warming of the surface temperatures. We focus first on the annual mean response, as it is the most relevant for changes in sea-level rise. In the All ensemble the annual mean SMB increases by 191.6

$Gtyr^{-1}$ over the second half of the 20th century (red bar), and ODS and stratospheric ozone make comparable contributions to the annual mean Antarctic SMB increase. Validating the results of Previdi and Polvani (2017) with a different model, we find that the increased emissions of ODS account for $20\%$ ($38.4\ Gtyr^{-1}$) of the annual mean SMB increase (blue bar); stratospheric ozone depletion accounts for $24.2\%$ ($46.4\ Gtyr^{-1}$) of the annual mean SMB increase (green bar). Tropospheric ozone, on the other hand, shows a statistically insignificant contribution of only $12.7\%$ ($24.5\ Gtyr^{-1}$); increases in tropospheric ozone are associated with air pollution, and thus are mostly concentrated in the Northern Hemisphere. Thus, ODS and stratospheric ozone together account for $\sim 44\%$ of the annual mean Antarctic SMB increase over the period of interest; the other $\sim 56\%$ of the increase is caused by other forcings dominated by increases in $CO_2$.

To further investigate the effects of ODS and ozone depletion on the annual mean SMB, we next examine the spatial pattern of $\Delta$SMB (Fig. 2). In the All ensemble the SMB increase mostly occurs in the Antarctic coastline, and over nearly all longitudes (green colors in Fig. 2a). While ODS (Fig. 2b) and tropospheric ozone (Fig. 2d) mostly increase the SMB over West Antarctica, stratospheric ozone increases it over East Antarctica and the Antarctic Peninsula (Fig. 2c). Thus, while ODS and stratospheric ozone have comparable impacts on the annual mean area integrated $\Delta$SMB (Fig. 1), it is the stratospheric ozone that is mostly responsible for the circumpolar increase of the SMB over the second half of the 20th century (Lenaerts et al., 2018).

Being powerful greenhouse gases, ODS increase the SMB by warming and moistening the Antarctic atmosphere, thus allowing for more snowfall (Previdi and Polvani, 2017). However, the reason for the SMB increase due to stratospheric ozone is less clear, and thus we next elucidate the underlying mechanism. Given the strong seasonal signal of ozone depletion, one would expect that its effects on the SMB would peak during Austral summer (December-February, DJF); the ozone hole itself initially develops during spring, but its tropospheric impacts are delayed until summer (Previdi and Polvani, 2014). Fig. 1b and c show $\Delta$SMB during DJF and June-August (JJA). Indeed in the All ensemble (red bars) the largest increase in SMB of $295.5\ Gtyr^{-1}$ occurs in DJF, compared with an increase of $124.1\ Gtyr^{-1}$ in JJA. Unlike the annual mean case, in DJF stratospheric ozone alone accounts for the majority ($63.2\%$) of the total SMB increase, and is more important than $CO_2$. In comparison, ODS (blue bar) and tropospheric ozone (purple bar) have only minor effects on the DJF $\Delta$SMB.

## 4   Elucidating the effects of stratospheric ozone on Antarctic SMB

In this section we explore the mechanism by which ozone depletion increases the Antarctic SMB, focusing our analysis on the DJF season when the ozone signal is strongest. Recently, Lenaerts et al. (2018) suggested that ozone depletion acts to increase, across most longitudes, the meridional geopotential height gradient at 500 hPa (associated with the zonal geostrophic wind), which they claimed was responsible for the circumpolar increase in DJF Antarctic SMB. This conclusion was based on the ensemble mean difference between the same full forcing (All) and Fix-O3 ensembles that are considered in the current study. To examine the robustness of that conclusion we follow Lenaerts et al. (2018), and show in Fig. 3 the difference in 500 hPa geopotential height (contours) and SMB (colors) between the All and the Fix-O3 experiments over the period 1986-2005 for each of the eight different ensemble members, and for the ensemble mean.

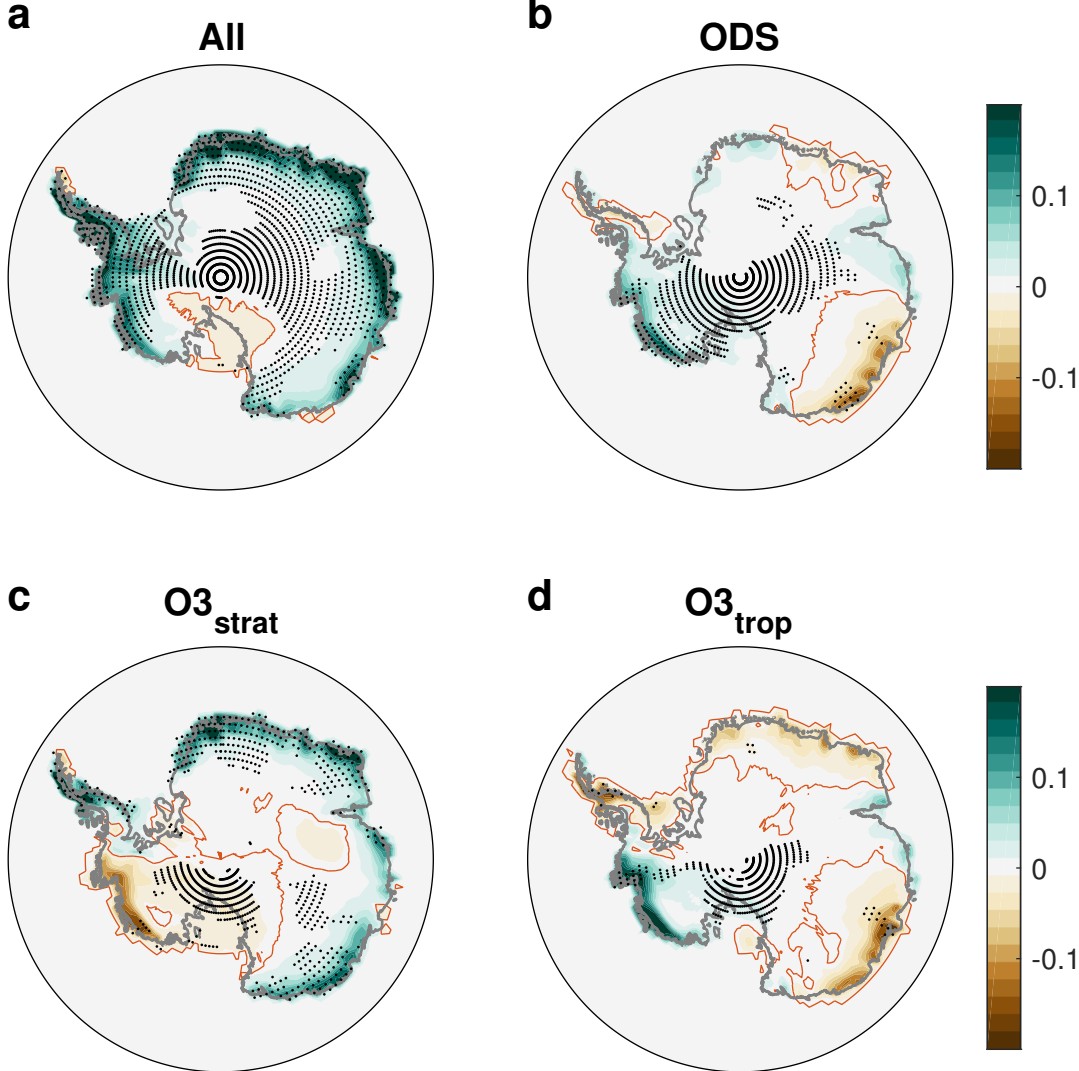

**Figure 2.** (a) Annual mean Antarctic SMB response to second half of the 20th century forcing ($\Delta$SMB, $\mathrm{Gtyr}^{-1}$) in the full forcing ensemble mean (All). Panels (b)-(d) show the contributions of ODS, stratospheric ozone ($\mathrm{O3_{strat}}$), and tropospheric ozone ($\mathrm{O3_{trop}}$) to the SMB response, respectively. Solid red lines mark the zero line. The small black dots show where the response is statistically significant at the 95% confidence level.

As in Lenaerts et al. (2018) (cf. their Fig. 2), the ensemble mean indeed shows an increase in the meridional geopotential height gradient and SMB across most longitudes. However, while all members show an increase in SMB around the Antarctic continent, only half of the members (#3,4,6 and 7) show a circumpolar increase in the meridional geopotential height gradient. This suggests that changes in the spatial patterns of the zonal geostrophic wind at 500 hPa are not robust across ensemble

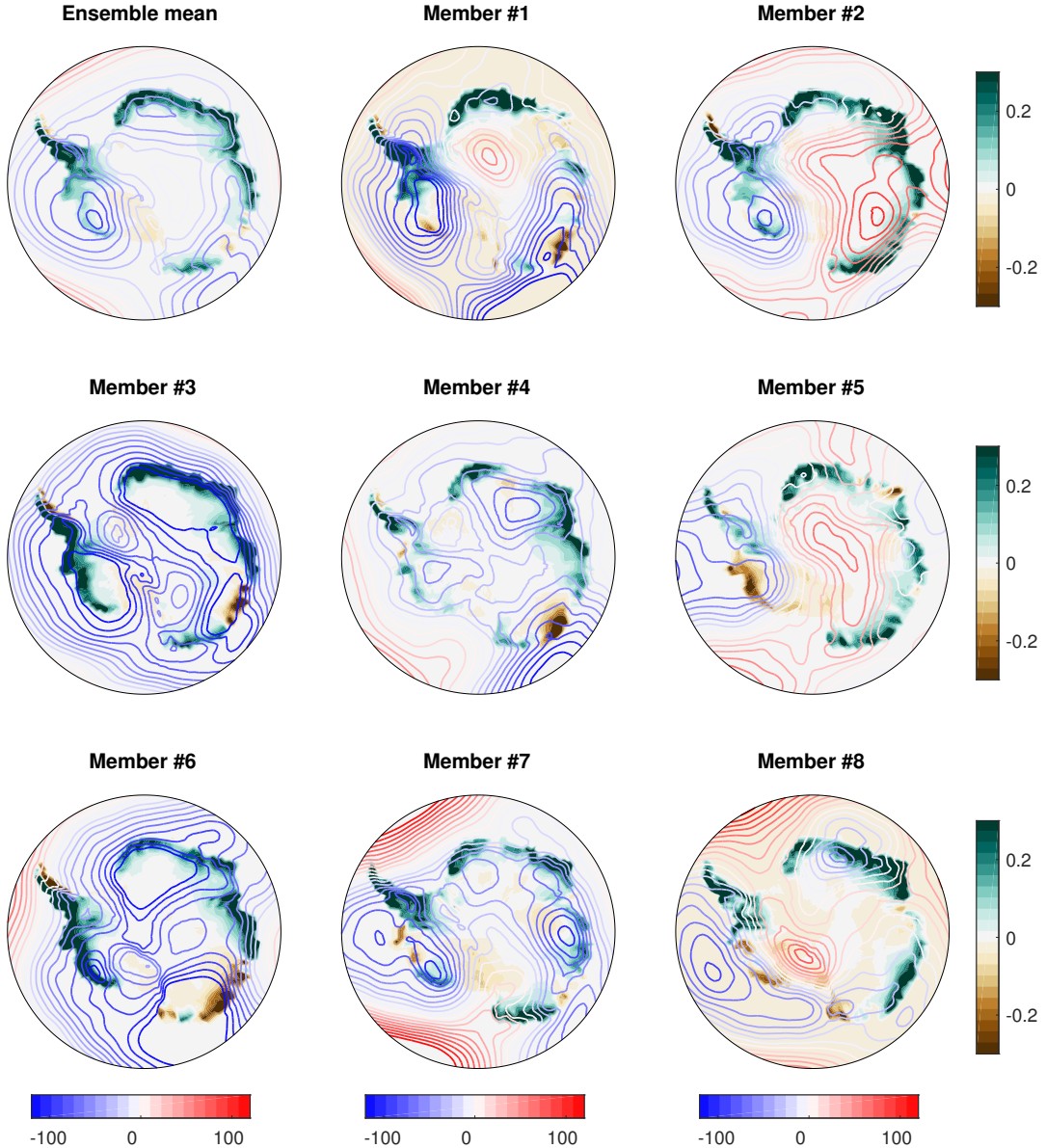

**Figure 3.** The difference in DJF Antarctic SMB (shading, $\mathrm{Gt\,yr^{-1}}$) and 500 hPa geopotential height (contours, $\mathrm{m^2s^{-2}}$) between the full forcing ensemble (All) and the fixed stratospheric and tropospheric ozone ensemble (Fix-O3), averaged over the period 1986-2005.

members. In any case, it is changes in the meridional convergence of moisture flux - and not changes in the mean zonal flow - that directly affect the zonal mean SMB (as shown below). We examine these meridional moisture flux changes next.

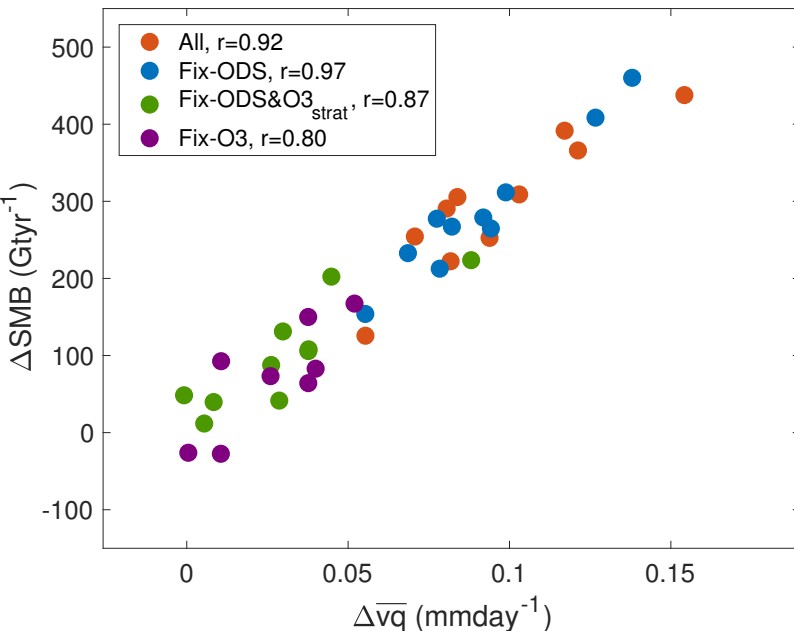

**Figure 4.** The response over the second half of the 20th century forcing of the DJF SMB ($\Delta$SMB) as a function of the meridional moisture flux convergence ($\Delta\overline{vq}$) in the full forcing ensemble (All, red), fixed ODS ensemble (Fix-ODS, blue), fixed ODS and stratospheric ozone ensemble (Fix–ODS&O3$_{\text{strat}}$, green) and fixed stratospheric and tropospheric ozone ensemble (Fix-O3, purple). Correlations appear at the upper left corner.

Changes in the surface precipitation-minus-evaporation (P-E; equivalent to the SMB over Antarctica) can be analyzed using the zonal mean vertically integrated moisture equation (e.g., Trenberth and Guillemot, 1995; Seager et al., 2010),

$$\Delta\left(\overline{P}(\phi) - \overline{E}(\phi)\right) = -\frac{1}{ga\cos\phi}\frac{\partial}{\partial\phi}\Big\{\int_{o}^{p_s}\left([\overline{v}]\Delta\overline{q} + [\overline{q}]\Delta\overline{v} + \Delta\overline{v'q'}\right)\cos\phi\,dp\Big\}, \tag{1}$$

where over bar represents zonal and DJF means, $g$ is gravity, $a$ is Earth's radius, $\phi$ is latitude, $p_s$ is surface pressure, $v$ is the meridional velocity, $q$ is specific humidity, square brackets represent time mean calculated across the combined 1990-2005 and 1940-1955 periods, and prime represents deviation from zonal and monthly mean (i.e., transient and stationary eddies). The first term on the righthand side of Eq. 1 accounts for changes in the mean moisture ($\overline{v}\Delta\overline{q}$), the second for the changes in meridional velocity ($\overline{q}\Delta\overline{v}$), and the third term accounts for changes in eddy moisture flux ($\Delta\overline{v'q'}$). The convergence of each

term on the righthand side is calculated over all latitudes. Note that $\overline{P} - \overline{E}$ is not identical to SMB (zonal mean $P - E$ vs area integrated $P - E$ over only the Antarctic continent), as it accounts for processes that occurs over the ocean (subantarctic regions accounts for a large portion of the Southern Ocean storms). To minimize this difference here we average each term in Eq. 1 between $65°\text{S} - 90°\text{S}$ to account for changes in $\overline{P} - \overline{E}$ mostly over the Antarctic continent. As a result, while changes in the total meridional moisture flux convergence ($\Delta\overline{vq}$, i.e., the sum of the right hand side terms in Eq. 1) are larger in magnitude

than $\Delta$SMB, they are highly correlated with $\Delta$SMB as one can see in Fig. 4 ($r = 0.92$ in All, $r = 0.97$ in Fix-ODS, $r = 0.87$

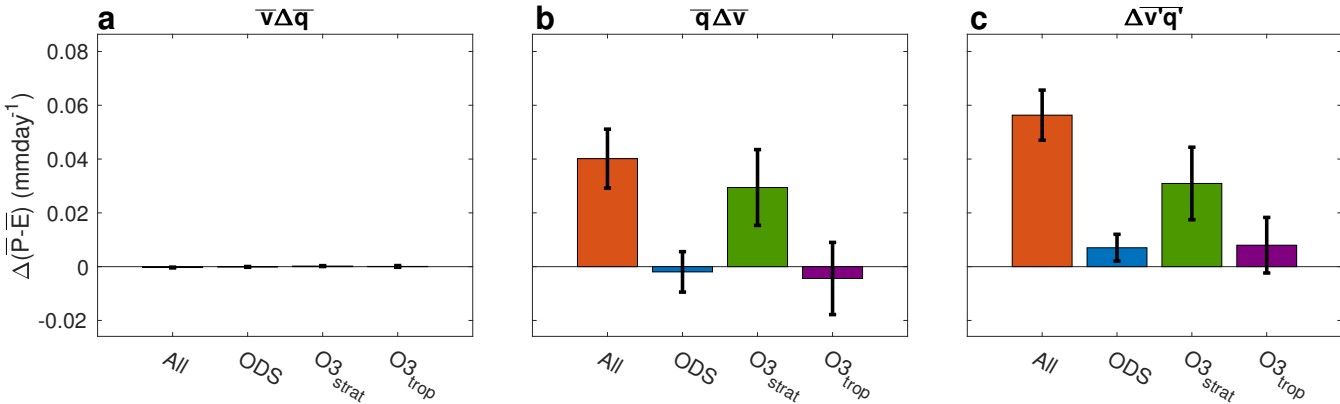

**Figure 5.** The contributions to the increase in DJF Antarctic $\overline{P} - \overline{E}$, $\Delta\left(\overline{P} - \overline{E}\right)$, arising from changes in (a) mean moisture ($\overline{v}\Delta\overline{q}$), (b) mean meridional circulation ($\overline{q}\Delta\overline{v}$) and (c) eddy moisture flux ($\Delta\overline{v'q'}$). Red bars show the response in the full forcing ensemble mean. The blue, green and purple bars show the contributions of ODS, stratospheric ozone (O3$_\text{strat}$), and tropospheric ozone (O3$_\text{trop}$) to the P-E response, respectively. Error bars represent the 95% confidence interval.

in Fix-ODS&O3$_\text{strat}$ and $r = 0.80$ in Fix-O3). The strong correlation between $\Delta\overline{vq}$ and $\Delta$SMB suggests that the zonal mean moisture budget (i.e., Eq. 1) can provide meaningful insight into the physical processes driving Antarctic SMB changes ($\overline{P} - \overline{E}$ processes over the ocean are linearly related to $\overline{P} - \overline{E}$ processes over land).

Figure 5 shows the contribution of each term in Eq. 1 to changes in DJF Antarctic $\overline{P} - \overline{E}$. First we focus on the response to all forcings (red bars). Moisture changes in DJF have only a minor effect on $\Delta\left(\overline{P} - \overline{E}\right)$ ($-0.03 \cdot 10^{-2}$ mmday$^{-1}$, red bar in Fig. 5a), in contrast to changes in the mean meridional velocity ($4 \cdot 10^{-2}$ mmday$^{-1}$, Fig. 5b) and eddy moisture flux ($5.6 \cdot 10^{-2}$ mmday$^{-1}$, Fig. 5c), which have a comparatively large effect. Second, similar to DJF $\Delta$SMB (Fig. 1b), ODS (blue bars) and tropospheric ozone (purple bars) have minor effects on $\Delta\left(\overline{P} - \overline{E}\right)$, in contrast to stratospheric ozone which has a relatively large effect (green bars). Specifically, ozone depletion is found to account for $73.3\%$ ($2.9 \cdot 10^{-2}$ mmday$^{-1}$) and $54.9\%$ ($3.1 \cdot 10^{-2}$ mmday$^{-1}$) of the mean meridional velocity and eddy moisture flux contributions to the total in DJF, respectively. This corroborates the results of previous studies who showed the important and increasing role of eddies, driven by ozone depletion, in converging more moisture into the Antarctic (Papritz et al., 2014; Grieger et al., 2018), and thus, as shown here, increasing the SMB by enhancing snowfall.

Finally we ask: how does ozone depletion enhance the meridional flow? Previous studies have shown that by increasing the meridional temperature gradient aloft, ozone depletion acts to enhance the mean zonal wind on the poleward flank of the jet (e.g., Polvani et al., 2011b). This enhanced zonal wind is not confined to the upper levels but reaches all the way to the surface. Fig. 6 shows the response in DJF mean zonal wind ($\Delta\overline{u}$) to the various forcings. In the historical (All) integrations, the mean zonal wind intensifies over the poleward flank of the jet (panel a), and this response is driven almost entirely by stratospheric ozone depletion (Fig. 6c).

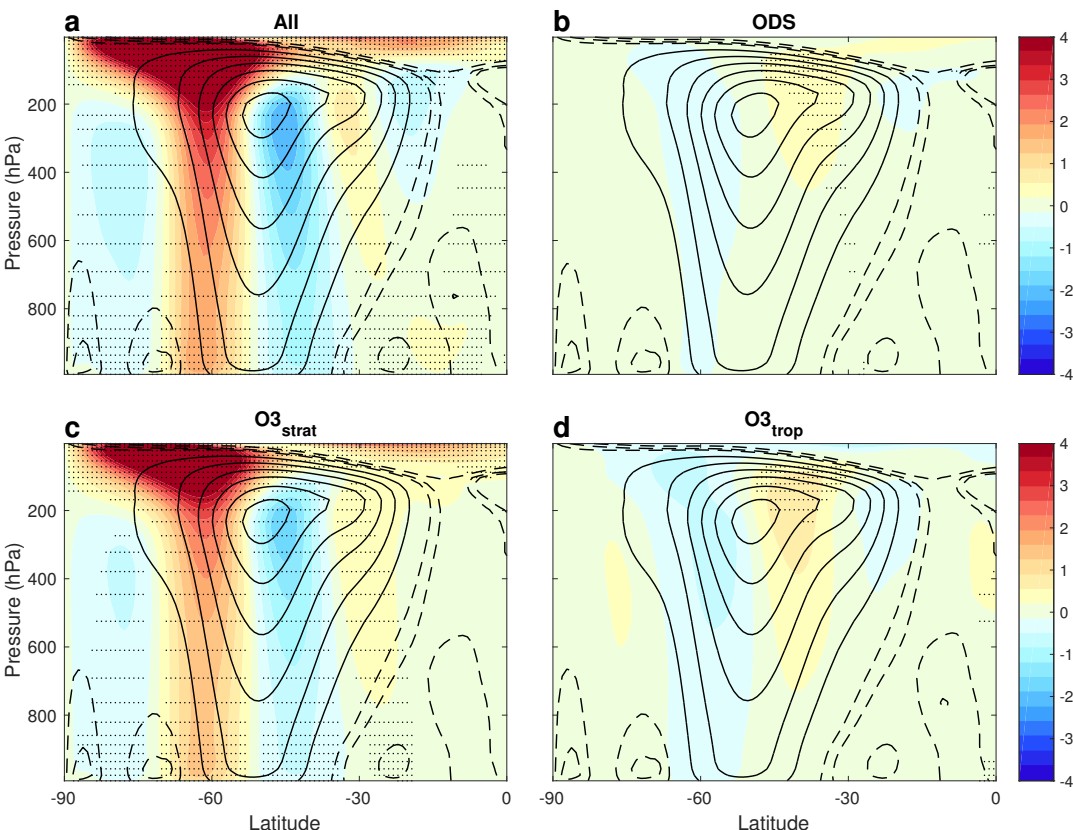

**Figure 6.** (a) The response over the second half of the 20th century forcing of the DJF zonal mean zonal wind ($\Delta\overline{u}$, ms$^{-1}$) in the All forcing ensemble mean. Panels (b)-(d) show the contributions of ODS, stratospheric ozone (O3$_{\text{strat}}$), and tropospheric ozone (O3$_{\text{trop}}$) to $\Delta\overline{u}$, respectively. Contours show the zonal mean zonal wind in the 1940-1955 period. The small black dots show where the response is statistically significant at the 95% confidence level.

145  Do the changes in the mean zonal wind due to ozone depletion imply an increase in the meridional flow? In the lower troposphere at mid-to-high latitudes the frictional force on the zonal wind balances the Coriolis force on the meridional flow, $r\overline{u} \approx f\overline{v}$, where $r$ is a drag constant and $f$ is the Coriolis parameter (cf. page 480 in Vallis, 2006). Note that eddy momentum fluxes do not appear in this balance as they are concentrated in the upper troposphere. This balance thus provides a link between ozone depletion and the meridional flow: the enhanced zonal wind at lower levels due to ozone depletion must be accompanied

150 by an increase in the meridional wind as well. To demonstrate this balance, Fig. 7 shows the correlation between the DJF $\Delta f\overline{v}$ and $\Delta\overline{u}$ averaged over the lower troposphere (600 hPa to surface) and over mid-high latitudes ($50°\text{S} - 70°\text{S}$) across the four ensembles. Changes in $f\overline{v}$ are very highly correlated with changes in $\overline{u}$, with $r = 0.99$ in the All ensemble (red dots), $r = 0.98$ for Fix-ODS (blue dots), $r = 0.99$ for Fix-ODS&O3$_{\text{strat}}$ (green dots) and $r = 0.94$ for Fix-O3 (purple dots). Not only is there an excellent correlation between $\Delta f\overline{v}$ and $\Delta\overline{u}$, but ensembles with ozone depletion (All and Fix-ODS) show a larger increase

155 in the mean zonal and meridional winds, in contrast to ensembles with fixed stratospheric ozone (Fix-ODS&O3$_{\text{strat}}$ and Fix-

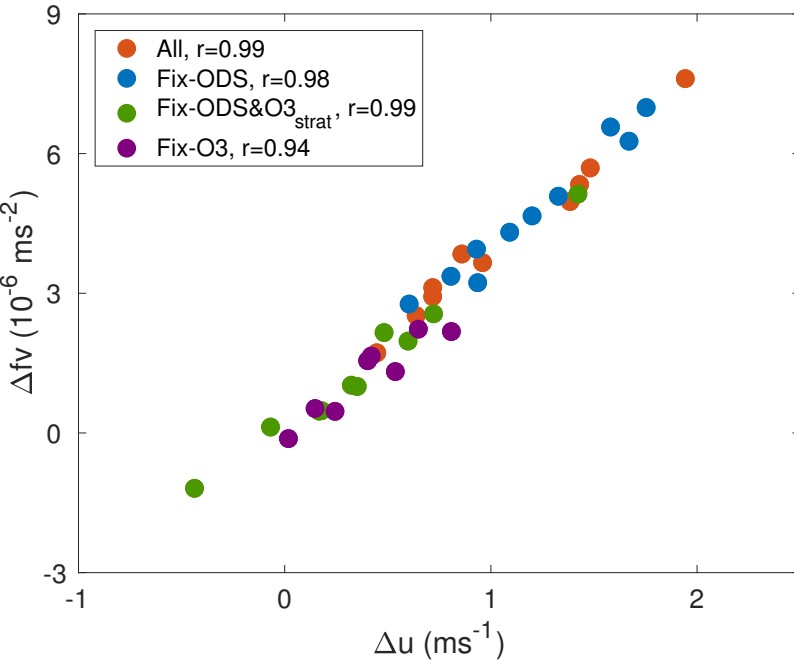

**Figure 7.** The response over the second half of the 20th century forcing of the DJF lower troposphere mid-high latitude Coriolis force on the meridional flow ($\Delta fv$) as a function of the mean zonal wind ($\Delta u$) in the full forcing ensemble (All, red), fixed ODS ensemble (Fix-ODS, blue), fixed ODS and stratospheric ozone ensemble (Fix-ODS&O3$_{strat}$, green) and fixed stratospheric and tropospheric ozone ensemble (Fix-O3, purple). Correlations appear at the upper left corner.

O3). Thus, the enhanced meridional flow in the All ensemble, and the associated increase in $\overline{P} - \overline{E}$, are largely due to the depletion of stratospheric ozone.

Changes in the mean zonal wind not only explain the increase in the mean meridional wind, but can be directly linked to the increase in the eddy moisture flux. Midlatitude eddies are driven by baroclinic instability (which arises from the vertical shear of the zonal wind, and the accompanying meridional temperature gradient), and thus the stronger increase of the mid-high latitudes ($50°S - 70°S$) mean zonal wind due to ozone depletion at upper levels relative to lower levels (i.e., an increase in the vertical shear, Fig. 6) suggests a strengthening of the eddies. However, during summer, the weak meridional temperature gradient might be insufficient to excite baroclinic eddies, and thus barotropic instability (which stems from the relative vorticity gradient, $u_{yy}$, where the y subscript represents the meridional derivative) might also be a driver of the midlatitude eddies.

To examine which instability drives the increase in eddy moisture flux, in Fig. 8 we show the correlation between the vertically integrated $\Delta \overline{v'q'}$ (the eddy moisture flux) and $\Delta \overline{u_z}$ (the vertical shear of the zonal wind, panel a) and the vertically averaged (through the entire atmosphere) $\Delta \overline{u_{yy}}$ (the curvature of the zonal winds, which is associated with barotropic instability, panel b). For simplicity, we define $\overline{u_z}$ as the zonal wind difference between upper (300 hPa-500 hPa) and lower

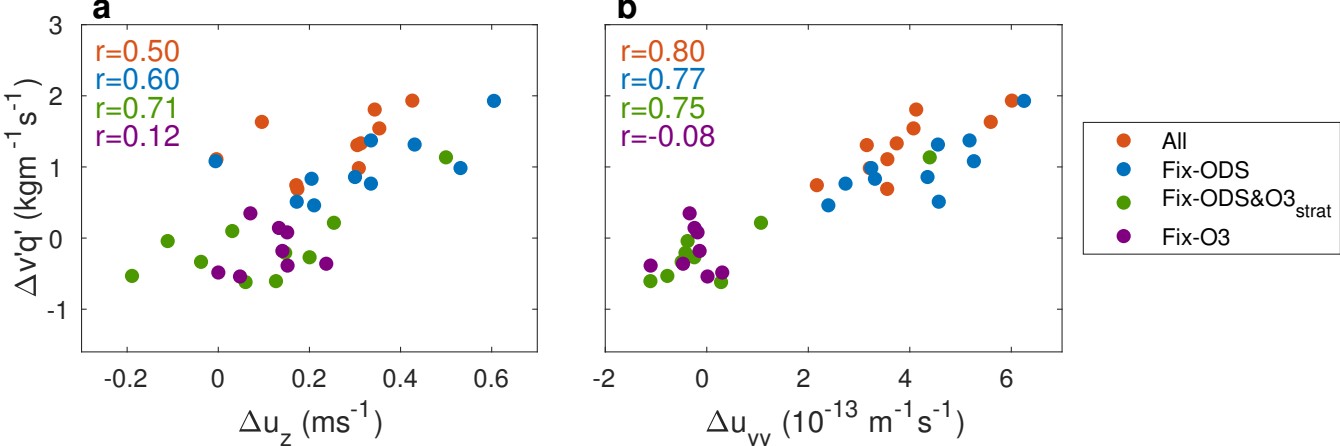

**Figure 8.** The response over the second half of the 20th century forcing of the DJF mid-high latitude eddy moisture flux ($\Delta\overline{v'q'}$) as a function of (a) the vertical shear of the zonal wind ($\Delta u_z$), and (b) the meridional gradient of the mean relative vorticity ($\Delta u_{yy}$) in the full forcing ensemble (All, red), fixed ODS ensemble (Fix-ODS, blue), fixed ODS and stratospheric ozone ensemble (Fix-ODS&O3$_{\text{strat}}$, green) and fixed stratospheric and tropospheric ozone ensemble (Fix-O3, purple). Correlations appear at the upper left corner of each panel.

(600 hPa-850 hPa) levels. All variables are averaged over mid-high latitudes ($50°S - 70°S$). We use here the absolute value of
$\overline{v'q'}$ so that the positive values of $\Delta\overline{v'q'}$ indicate strengthening of the eddy moisture flux onto the Antarctic continent.

As seen in Fig. 8, $\Delta\overline{v'q'}$ has a modest correlation with $\Delta\overline{u_z}$ across most ensembles ($r = 0.50$ in All, $r = 0.60$ in Fix-ODS, $r = 0.71$ in Fix-ODS&O3$_{\text{strat}}$ and $r = 0.12$ in Fix-O3). More importantly, ensembles with ozone depletion (red and blue dots) do not show a significantly larger increase in $\overline{u_z}$ relative to ensembles with fixed stratospheric ozone (purple and green dots): ozone depletion thus has a weak impact on DJF $\overline{u_z}$. In contrast, $\Delta\overline{v'q'}$ has a higher correlation with $\Delta\overline{u_{yy}}$ across most
ensembles ($r = 0.80$ in All, $r = 0.77$ in Fix-ODS, $r = 0.75$ in Fix-ODS&O3$_{\text{strat}}$ and $r = -0.08$ in Fix-O3), and ensembles with ozone depletion (red and blue dots) do show a significant increase in $\overline{u_{yy}}$ relative to ensembles with fixed stratospheric ozone (purple and green dots). This analysis suggests that increased barotropic instability is the primary mechanism via which ozone depletion enhances the eddy-moisture flux, resulting in a larger Antarctic SMB over the second half of the 20th century.

## 5   Conclusions

Two recent studies have suggested that increasing ozone depleting substances (ODS) and the accompanying loss of stratospheric ozone have caused a substantial fraction of the increase in Antarctic surface mass balance over the second half of the 20th century. Neither study, however, cleanly separated these forcings. We here quantified the separate contribution of these forcing agents in increasing the Antarctic SMB, using fixed-forcing ensembles of model simulations. Our results show that ODS and stratospheric ozone have had comparable effects on the annual mean Antarctic SMB, and together account

for $\sim 40\%$ of the SMB increase over the second half of the 20th century. The effect of stratospheric ozone are especially pronounced during Austral summer, when they account for $\sim 60\%$ of the SMB increase.

We have also shown that ozone depletion affects the SMB by enhancing the meridional circulation (mean and eddies), thus converging more water vapor over the Antarctic continent, and leading to increases in snowfall. The enhanced meridional flow is linked to ozone depletion through changes in the mean zonal wind. Specifically, in the lower troposphere, a stronger mean

zonal wind at mid-high latitudes is balanced by a stronger mean meridional wind. Additionally, increases in the meridional gradient of the zonal mean relative vorticity, due to ozone-induced zonal wind changes, enhances barotropic instability, and leads to increases in meridional eddy moisture fluxes.

Our results have confirmed that ODS and the accompanying depletion of stratospheric ozone have substantially contributed to the recent increases in Antarctic SMB and, therefore, the phase out of ODS by the Montreal Protocol, and the accompanying

recovery of stratospheric ozone, will act to decrease the SMB over the next several decades. The effect of ODS reduction and ozone recovery on the SMB will thus oppose the effect of increasing greenhouse gases, particularly during Austral summer. This has implications for the emergence/identification of SMB increases in observations, i.e., not only could this emergence be delayed (or masked) by natural variability (Previdi and Polvani, 2016), but will also be delayed as a consequence of the Montreal Protocol (Polvani et al., 2011a; Barnes et al., 2014).

*Code and data availability.* Data and codes are available upon request from: rc3101@colubia.edu

*Author contributions.* RC analyzed the data, MRE conducted the runs, and together with MP and LMP discussed and wrote the manuscript.

*Competing interests.* The authors declare that they have no conflict of interest.

*Acknowledgements.* We acknowledge the CESM Large Ensemble Community Project and supercomputing resources provided by NSF/CISL/Yellowstone. This research is founded by a grant from the National Science Foundation to Columbia University.

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
