# Peer review of "Distinguishing the impacts of ozone and ozone depleting substances on the recent increase in Antarctic surface mass balance"

_The Cryosphere, 2020_

## Referee Comment (RC1) · Ryan Fogt (Referee) · 29 Jul 2020

Review of 'Distinguishing the impacts of ozone and ozone depleting substances on the recent increase in Antarctic surface mass balance' by Chemke et al.

Overview: This paper used specialized climate model simulations from CESM to analyze the relative contribution of ozone depleting substances, and stratospheric and tropospheric ozone (separately) on changes in Antarctic mass balance. The study clearly demonstrates and cleanly separates that the largest contributions come from stratospheric ozone in austral summer. This is accomplished through changes in the meridional moisture flux, strongly tied to barotropic instability (rather than baroclinicity)

[Figure]

bringing more moisture to the Antarctic continent and increasing SMB.

The paper is well-written, the figures are clear, and the results are fully justified by the analysis. I only offer one small potential minor revision to help place the paper in a broader context of the model reliability, which never really was addressed or referenced. It would be helpful to know that the values and changes of SMB are well within the known bounds of SMB from satellite observations (surface height estimates etc.) and other detailed models of SMB.

Minor revision suggestion: 1. There is never really a discussion on how well the models employed do at simulating observed Antarctic SMB from satellite measurements or in comparison to more sophisticated models of SMB. At the very least, the LENS simulations could be compared to this over a period of overlap.

Specific technical edits: 2. Throughout: east Antarctica, west Antarctica, and Antarctic peninsula can all be capitalized since they refer to specific proper nouns / geographic regions: East Antarctica, West Antarctica, Antarctic Peninsula 3. Line 175 – change 'show' to 'shown' 4. Some of the nomenclature is a bit awkward, particularly in Fig. 8, why not just use derivatives instead of subscripts?

---

## Referee Comment (RC2) · Anonymous Referee #2 · 19 Aug 2020

The submission has the potential to make a significant contribution to the literature, but it is not quite there yet. Before I would be able to recommend acceptance, there are a number of issues which need to be addressed.

Lines 12-35: Some of this literature is quite dated. In these various aspects here also make reference to the following more recent investigations – Golledge N. R. (2020) Long-term projections of sea-level rise from ice sheets. Wiley Interdisciplinary Reviews-Climate Change 11, e634, doi: 10.1002/wcc.634. Eric Rignot, Jeremie Mouginot, Bernd Scheuchl, Michiel van den Broeke, Melchior J. van Wessem and Mathieu Morlighem, 2019: Four decades of Antarctic Ice Sheet mass balance from 1979-

2017. Proceedings of the National Academy of Sciences of the United States of America, 116, 1095-1103, doi: 10.1073/pnas.1812883116. Cecile Agosta, Charles Amory, Christoph Kittel, Anais Orsi, Vincent Favier, Hubert Gallee, Michiel R. van den Broeke, Jan T. M. Lenaerts, Jan Melchior van Wessem, Willem Jan van de Berg and Xavier Fettweis, 2019: Estimation of the Antarctic surface mass balance using the regional climate model MAR (1979-2015) and identification of dominant processes. Cryosphere, 13, 281-296, doi: 10.5194/tc-13-281-2019.

Lines 24-31: Ozone depletion and associated changes in subantarctic synoptic activity have been linked with variations in the polar transport of a wide range of atmospheric constituents. To point to this broader context beneficial to cite the paper of Cataldo, M., H. Evangelista, et al., 2013: Mineral dust variability in central West Antarctica associated with ozone depletion. Atmos. Chem. Phys., 13, 2165-2175.

Lines 41-42: The paper should be clearer in describing the initial conditions (and specifically that temperature) for the different ensemble members, or this aspect should be deleted. The comments presented here are not particularly clear. Having said that, the relevant text from Jennifer Kay's 2015 CESM paper (cited here) is also somewhat unclear in saying '... spread in ensemble members 3–30 was generated by round-off level differences in their initial air temperature fields. Specifically, we applied random round-off level (order of $10^{**}{-14}$ K) differences to the air temperature field of ensemble member 1 to generate atmospheric initial conditions for ensemble members 3–30' (page 1337). More detail is required. E.g., was the perturbation applied independently to each grid point? Were the affected points spread over the global 3D model space, or restricted to certain models levels and/or geographical regions.

Lines 51-55: The analysis and results presented here are contingent on this linear assumption. This warrants some more words as to what the pitfalls and biases (e.g., introduced by self-dampening or self-amplifying processes) might be. Would be helpful here to reference some of the relevant comments in the LARMIP V2 paper of Anders Levermann, Ricarda Winkelmann, Torsten Albrecht, . . . and Roderik S. W. van de Wal,

2020: Projecting Antarctica's contribution to future sea level rise from basal ice shelf melt using linear response functions of 16 ice sheet models (LARMIP-2). Earth System Dynamics, 11, 35-76, doi: 10.5194/esd-11-35-2020.

Lines 62-63: Delete parenthetical comment. Acronym SMB has essential already been defined (at line 15).

Line 72 (Figure 2): Valuable and informative plots here for annual case. Might be useful to plot the 'zero line' on these. The plots show a great deal of white, but one is not sure what parts are showing increases versus decreases. Regarding the statistical significance of the changes, the panels for the three forcing experiments (b – d) show, e.g., significant changes centered on the 180E meridian from the RIS to the pole, while no such significance is displayed in the 'All' case. This may be OK (I haven't thought this thru) but at first sight looks strange, and is worth checking. It may be that the values are negative in the 'O3 strat' case (while they are unambiguously positive I panels b and d. Showing the zero line would help with this possible riddle.

Line 93: Replace 'mb' with 'hPa'. (Similar comment throughout the Ms, including lines 84, 96, 102, 139, 156, . . .)

Lines 105-110: In equn (1) use extra nested parentheses to make clear that the vertical integral is taken over all terms in the integrand and not just the last one. Also, the RHS of the equation gives the (mathematical) convergence of moisture at latitude phi. This latitude dependence must be indicated on the LHS, as well as what infinitesimal area is being considered here. (The flux convergence relationship with P-E only make sense when a finite area is being considered.) Appropriate overbars are required on symbols in the legends in Figure 5, as well as in the caption.

Lines 112-124: Some of this material is poorly expressed and misleading. The net flux of moisture into the polar 'cap' (poleward of 65S), can actually be determined directly from the flux across 65S. The authors comment that changes in this net flux can be used to account for changes in SMB mostly over the Antarctic continent. This statement

is very misleading. The seas around Antarctic are host to intense and frequent storms (make reference here to compilation of Keay et al., 2003: Synoptic activity in the seas around Antarctica. Mon. Wea. Rev., 131, 272-288). These are associated with large P-E in those subantarctic waters (and certainly south of 65S). Hence, a priori, the net precip south of that latitude cannot be regarded as a proxy for SMB over the content itself. In justifying this association, the authors cite the correlation of these two P-Es, but do not compare their MAGNITUDES. A quick calculation reveals these to be very different. The 65S polar cap covers about 25 million km**2. A flux of 0.1 mm/day (a typical value in Figure 4) gives a volume of water of 10**-4 m/day * 25 x 10**12 m**2 = 2.5 x 10**9 m**3/day. This is 2.5 Gt/day or 910 Gt/yr. This is about 3 times larger than the 300 Gt/yr (for 0.1 mm/day) indicated in Figure 4. This means that about two thirds of the vapor which crosses 65S precipitates into the ocean before it reaches the continent. While one MIGHT expect that if the total flux 65S changes by a certain fraction the P-E over the continent would change by a similar fraction. However this important part of the text must be expressed and justified more clearly. I appreciate that calculating eddy fluxes across latitudes in much easier than across irregular boundaries (like the Antarctic coast), and I have no great problem with what the authors have done. My main point is that they should be much more upfront with the caveats, and be clear on the synoptics in this complex part of the world.

Lines 118-124: Reinforce this message by refencing the study of Grieger, Leckebusch, et al., 2018: Subantarctic cyclones identified by 14 tracking methods, and their role for moisture transports into the continent. Tellus, 70A, 1454808, doi: 10.1080/16000870.2018.1454808 which demonstrates strong positive summer trends in subantarctic cyclone numbers (in association in increasing SAM), and their role in poleward moisture transport. Also valuable here to cite here the analysis of Papritz, Wernli et al. 2014: The role of extratropical cyclones and fronts for Southern Ocean freshwater fluxes. J. Clim., 27, 6205-6224 exploring the SH relationships between synoptic eddies and P-E.

Lines 153-166: Some interesting conclusions are reached here in connection with the relative importance of baroclinic and barotropic instability in driving the moisture-transporting transient eddies. I have a few issues with how this comparison was made. For the baroclinic part the authors calculate the delta vertical wind shear as the zonal wind difference between upper (300mb-500mb) and lower (600mb-800mb) levels. This will be very similar to the difference between 400 and 700 hPa, or an estimate of the baroclinicity at 550 hPa. The total moisture transport is dominated by the lower levels of the troposphere, and the 850 hPa level (which is frequently chosen for applications such as this) would be a much more appropriate level to take. For the barotropic instability case, the text states the this is determined from the 'vertically averaged delta uyy'. It is not clear whether this average was taken through the entire atmosphere and/or why were not some key atmospheric levels chosen for this. At the very least the authors should determine their baroclinicity at a more physically-consistent level, and justify why a similar level should not be used for the barotropic component. As it stands, the analysis has not convinced me that it '. . . suggests that increased barotropic instability is the primary mechanism via which ozone depletion enhances the eddy-moisture flux, resulting in a larger Antarctic SMB over the second half of the 20th century.

Lines 179: After '. . . gradient of the mean relative vorticity' insert 'of the zonal flow'. (This connection has only been shown for uyy.)
* * *

---

## Author Comment (AC1) · 16 Sep 2020

Reply to reviewer comments:
Distinguishing the impacts of ozone and ozone depleting substances on the recent increase in Antarctic surface mass balance

Dear Dr. Savarino,

Thank you for considering this paper for publication in The Cryosphere. We have found the reviewer's comments insightful and helpful. Please find below a point-by-point reply (in black) to all the reviewers' comments (in blue).

**Reviewer 1**

Overview: This paper used specialized climate model simulations from CESM to analyze the relative contribution of ozone depleting substances, and stratospheric and tropospheric ozone (separately) on changes in Antarctic mass balance. The study clearly demonstrates and cleanly separates that the largest contributions come from stratospheric ozone in austral summer. This is accomplished through changes in the meridional moisture flux, strongly tied to barotropic instability (rather than baroclinicity) bringing more moisture to the Antarctic continent and increasing SMB. The paper is well-written, the figures are clear, and the results are fully justified by the analysis. I only offer one small potential minor revision to help place the paper in a broader context of the model reliability, which never really was addressed or referenced. It would be helpful to know that the values and changes of SMB are well within the known bounds of SMB from satellite observations (surface height estimates etc.) and other detailed models of SMB.

We thank the reviewer for the careful reading and very useful comments.

Minor revision suggestion: 1. There is never really a discussion on how well the models employed do at simulating observed Antarctic SMB from satellite measurements or in comparison to more sophisticated models of SMB. At the very least, the LENS simulations could be compared to this over a period of overlap.

Following the reviewer's comment we now discuss the ability of the CESM to capture the Antarctic SMB (lines 39-43). Previous studies have shown that the CESM captures the spatial patterns of climatological mean Antarctic SMB, and its variability, from ice-cores and reanalysis (Lenaerts et al., 2018). Furthermore, the CESM well captures the climate response in the Southern Hemisphere to ozone-depletion (England et al., 2016; Landrum et al., 2017). This provide us confidence in using the CESM to investigate SMB changes forced by different ozone related agents.

Specific technical edits: 2. Throughout: east Antarctica, west Antarctica, and Antarctic peninsula can all be capitalized since they refer to specific proper nouns / geographic regions: East Antarctica, West Antarctica, Antarctic Peninsula

We have capitalized all 'geographic regions' (lines 84-85).

3. Line 175 – change 'show' to 'shown'

Done (line 189).

4. Some of the nomenclature is a bit awkward, particularly in Fig. 8, why not just use derivatives instead of subscripts?

We would prefer to minimize the number of symbols on each figure, and to keep the current subscripts as 'derivatives'.

We would like to emphasize again our gratitude to the reviewer who pointed us in important directions that have significantly improved this manuscript.

Sincerely,

Rei Chemke, Michael Previdi, Mark England and Lorenzo Polvani

**References**

England, M. R., Polvani, L. M., Smith, K. L., Landrum, L., and Holland, M. M. (2016). Robust response of the Amundsen Sea Low to stratospheric ozone depletion. *Geophys. Res. Lett.*, 43(15):8207–8213.

Landrum, L. L., Holland, M. M., Raphael, M. N., and Polvani, L. M. (2017). Stratospheric Ozone Depletion: An Unlikely Driver of the Regional Trends in Antarctic Sea Ice in Austral Fall in the Late Twentieth Century. *Geophys. Res. Lett.*, 44(21):11,062–11,070.

Lenaerts, J. T. M., Fyke, J., and Medley, B. (2018). The Signature of Ozone Depletion in Recent Antarctic Precipitation Change: A Study With the Community Earth System Model. *Geophys. Res. Lett.*, 45(23):12,931–12,939.

---

## Author Comment (AC2) · 16 Sep 2020

Reply to reviewer comments:
Distinguishing the impacts of ozone and ozone depleting substances on the recent increase in Antarctic surface mass balance

Dear Dr. Savarino,

Thank you for considering this paper for publication in The Cryosphere. We have found the reviewer's comments insightful and helpful. Please find below a point-by-point reply (in black) to all the reviewers' comments (in blue).

**Reviewer 2**

The submission has the potential to make a significant contribution to the literature, but it is not quite there yet. Before I would be able to recommend acceptance, there are a number of issues which need to be addressed.

We thank the reviewer for the careful reading and very useful comments.

Lines 12-35: Some of this literature is quite dated. In these various aspects here also make reference to the following more recent investigations – Golledge N. R. (2020) Long-term projections of sea-level rise from ice sheets. Wiley Interdisciplinary Reviews-Climate Change 11, e634, doi: 10.1002/wcc.634. Eric Rignot, Jeremie Mouginot, Bernd Scheuchl, Michiel van den Broeke, Melchior J. van Wessem and Mathieu Morlighem, 2019: Four decades of Antarctic Ice Sheet mass balance from 1979-2017. Proceedings of the National Academy of Sciences of the United States of America, 116, 1095-1103, doi: 10.1073/pnas.1812883116. Cecile Agosta, Charles Amory, Christoph Kittel, Anais Orsi, Vincent Favier, Hubert Gallee, Michiel R. van den Broeke, Jan T. M. Lenaerts, Jan Melchior van Wessem, Willem Jan van de Berg and Xavier Fettweis, 2019: Estimation of the Antarctic surface mass balance using the regional climate model MAR (1979-2015) and identification of dominant processes. Cryosphere, 13, 281-296, doi: 10.5194/tc-13-281-2019.

We thank the reviewer for bringing these papers to our attention (Agosta et al., 2019; Rignot et al., 2019; Golledge, 2020), which are now cited in the above lines (lines 14-19). Please note that Agosta et al. (2019) discussed the climatology of Antarctic SMB and focuses on year 2015, while here (including in the introduction) we discuss the trends in SMB.

Lines 24-31: Ozone depletion and associated changes in subantarctic synoptic activity have been linked with variations in the polar transport of a wide range of atmospheric constituents. To point to this broader context beneficial to cite the paper of Cataldo, M., H. Evangelista, et al., 2013: Mineral dust variability in central West Antarctica associated with ozone depletion. Atmos. Chem. Phys., 13, 2165-2175.

We thank the reviewer for this comment, and added the above reference (Cataldo et al., 2013) to the text (line 25). Indeed the effects of Ozone depletion on the atmospheric flow have not only increased the Antarctic SMB, but also the incursion of atmospheric dust into the Antarctica continent.

Lines 41-42: The paper should be clearer in describing the initial conditions (and specifically that temperature) for the different ensemble members, or this aspect should be deleted. The comments presented here are not particularly clear. Having said that, the relevant text from Jennifer Kay's 2015 CESM paper (cited here) is also somewhat unclear in saying '... spread in ensemble members 3–30 was generated by round-off level differences in their initial air temperature fields. Specifically, we applied random round-off level (order of 10**–14 K) differences to the air temperature field of ensemble member 1 to generate atmospheric initial conditions for ensemble members 3–30' (page 1337). More detail is required. E.g., was the perturbation applied independently to each grid point? Were the affected points spread over the global 3D model space, or restricted to certain models levels and/or geographical regions.

Following the reviewer's comment we further explain the construction of the CESM large ensembles (lines 47-48). For the construction of the large-ensemble described in Kay et al. (2015) the first simulation (member) is initialized from a random year in the long preindustrial control run (under 1850 forcing), and runs from 1850 to 1920 under the Historical forcing. At 1920 the other members of the ensemble branch off the first member, and are initialized with a minor change in air temperature ($\mathcal{O}10^{-14}$K). This perturbation is presented in all levels and locations (all grid points).

Lines 51-55: The analysis and results presented here are contingent on this linear assumption. This warrants some more words as to what the pitfalls and biases (e.g., introduced by self-dampening or self-amplifing processes) might be. Would be helpful here to reference some of the relevant comments in the LARMIP V2 paper of Anders Levermann, Ricarda Winkelmann, Torsten Albrecht, . . . and Roderik S. W. van de Wal 2020: Projecting Antarctica's contribution to future sea level rise from basal ice shelf melt using linear response functions of 16 ice sheet models (LARMIP-2). Earth System Dynamics, 11, 35-76, doi: 10.5194/esd-11-35-2020.

Following the reviewer's comment we discuss what biases may arise from the assumption that the forcings are linearly additive (lines 61-63). As discussed in Levermann et al. (2020), by assuming additivity we neglect any self-dampening or self-amplifying processes of each forcing agent: our approach assumes that the SMB responses to different forcings are additive, and thus that the effects of non-linear interactions between forcings can be ignored. However, as discussed in Levermann et al. (2020), since we are investigating the SMB forced response to external forcings, rather its internal variability, we reduce these internal processes and increase the linearity of the results. Furthermore, the comparison between each individual forcing experiment and the 'All' experiment, allow us to assess whether such self-amplifying/decaying processes are not represented in the overall response.

Lines 62-63: Delete parenthetical comment. Acronym SMB has essential already been defined (at line 15).

Done.

Line 72 (Figure 2): Valuable and informative plots here for annual case. Might be useful to plot the 'zero line' on these. The plots show a great deal of white, but one is not sure what parts are showing increases versus decreases. Regarding the statistical significance of the changes, the panels for the three forcing experiments (b – d) show, e.g., significant changes centered on the 180E meridian from the RIS to the pole, while no such significance is displayed in the 'All' case. This may be OK (I haven't thought this thru) but at first sight looks strange, and is worth checking. It may be that the values are negative in the 'O3 strat' case (while they are unambiguously positive I panels b and d. Showing the zero line would help with this possible riddle.

Following the reviewer's comment we have further amplified the colors in Fig. 2 and added the zero lines. The great deal of white in these plots are a result of the large SMB changes in the Antarctic periphery, with minor changes inland. Note that 'All' includes not only the forcing agents that we explore here (ODS and ozone), but all other anthropogenic or natural forcing agents (e.g., greenhouse gases). Thus, not necessarily changes in ODS and ozone could explain the changes when accounting for all forcing agents.

Line 93: Replace 'mb' with 'hPa'. (Similar comment throughout the Ms, including lines 84, 96, 102, 139, 156, . . .)

Done.

Lines 105-110: In equn (1) use extra nested parentheses to make clear that the vertical integral is taken over all terms in the integrand and not just the last one. Also, the RHS of the equation gives the (mathematical) convergence of moisture at latitude phi. This latitude dependence must be indicated on the LHS, as well as what infinitesimal area is being considered here. (The flux convergence relationship with P-E only make sense when a finite area is being considered.) Appropriate overbars are required on symbols in the legends in Figure 5, as well as in the caption.

Following the reviewer's comment we added extra parentheses to Eq. 1, and the LHS dependence in latitude. The convergence is over the meridional grid cell length (lines 120-121), and we added overbars to Figs. 4 and 5.

Lines 112-124: Some of this material is poorly expressed and misleading. The net flux of moisture into the polar 'cap' (poleward of 65S), can actually be determined directly from the flux across 65S. The authors comment that changes in this net flux can be used to account for changes in SMB mostly over the Antarctic continent. This statement is very misleading. The seas around Antarctic are host to intense and frequent storms (make reference here to compilation of Keay et al., 2003: Synoptic activity in the seas around Antarctica. Mon. Wea. Rev., 131, 272-288). These are associated with large P-E in those

subantarctic waters (and certainly south of 65S). Hence, a priori, the net precip south of that latitude cannot be regarded as a proxy for SMB over the content itself. In justifying this association, the authors cite the correlation of these two P-Es, but do not compare their MAGNITUDES. A quick calculation reveals these to be very different. The 65S polar cap covers about 25 million km**2. A flux of 0.1 mm/day (a typical value in Figure 4) gives a volume of water of 10**-4 m/day * 25 x 10**12 m**2 = 2.5 x 10**9 m**3/day. This is 2.5 Gt/day or 910 Gt/yr. This is about 3 times larger than the 300 Gt/yr (for 0.1 mm/day) indicated in Figure 4. This means that about two thirds of the vapor which crosses 65S precipitates into the ocean before it reaches the continent. While one MIGHT expect that if the total flux 65S changes by a certain fraction the PE over the continent would change by a similar fraction. However this important part of the text must be expressed and justified more clearly. I appreciate that calculating eddy fluxes across latitudes in much easier than across irregular boundaries (like the Antarctic coast), and I have no great problem with what the authors have done. My main point is that they should be much more upfront with the caveats, and be clear on the synoptics in this complex part of the world.

Following the reviewer's comment we further explain the biases resulted from using zonal mean approach on a non-zonal problem (lines 121-130). When computing the moisture budget analysis we define eddies as deviation from zonal mean. Thus, the P-E problem only hold under zonal averaging. As the reviewer mentioned, the Antarctic coast line is not zonally symmetric, and thus our analysis indeed cover areas of water, and not only land as in the SMB calculation. The choice of 65S as the lower boundary for the analysis is twofold. On the one hand, the southern boundary should account for the Antarctic periphery where most of the SMB changes occur (Fig. 2 in the manuscript). On the other hand, the southern boundary should not be too far away towards the equator, in order to exclude open water areas as much as possible. Indeed, the magnitude of zonal mean P-E and SMB will not be identical (due to P-E processes over ocean; subantarctic regions accounts for a large portion of the Southern Ocean storms, Simmonds et al., 2003), however, the linear and high correlation between the two indicates that the processes over the ocean are linearly related to the process over land. Thus, changes in P-E could, to the very least, point us to which physical processes are mostly important in modifying the SMB.

Lines 118-124: Reinforce this message by refencing the study of Grieger, Leckebusch, et al., 2018: Subantarctic cyclones identified by 14 tracking methods, and their role for moisture transports into the continent. Tellus, 70A, 1454808, doi: 10.1080/16000870.2018.1454808 which demonstrates strong positive summer trends in subantarctic cyclone numbers (in association in increasing SAM), and their role in poleward moisture transport. Also valuable here to cite here the analysis of Papritz, Wernli et al. 2014: The role of extratropical cyclones and fronts for Southern Ocean freshwater fluxes. J. Clim., 27, 6205-6224 exploring the SH relationships between synoptic eddies and P-E.

Following the reviewer's comment we now cite the above studies (lines 138-140), which show the important and increasing role of eddies in converging more moisture water into the Antarctic (Papritz et al., 2014; Grieger et al., 2018).

Lines 153-166: Some interesting conclusions are reached here in connection with the relative importance of baroclinic and barotropic instability in driving the moisturetransporting transient eddies. I have a few issues with how this comparison was made. For the baroclinic part the authors calculate the delta vertical wind shear as the zonal wind difference between upper (300mb-500mb) and lower (600mb-800mb) levels. This will be very similar to the difference between 400 and 700 hPa, or an estimate of the baroclinicity at 550 hPa. The total moisture transport is dominated by the lower levels of the troposphere, and the 850 hPa level (which is frequently chosen for applications such as this) would be a much more appropriate level to take. For the barotropic instability case, the text states the this is determined from the 'vertically averaged delta uyy'. It is not clear whether this average was taken through the entire atmosphere and/or why were not some key atmospheric levels chosen for this. At the very least the authors should determine their baroclinicity at a more physically-consistent level, and justify why a similar level should not be used for the barotropic component. As it stands, the analysis has not convinced me that it '. . . suggests that increased barotropic instability is the primary mechanism via which ozone depletion enhances the eddy-moisture flux, resulting in a larger Antarctic SMB over the second half of the 20th century

Following the reviewer's comment we added the 850hPa level to the calculation of the vertical wind shear, and state that the barotropic component is vertically integrated through the entire atmosphere (lines 169 and 171). We would prefer to keep the baroclinic upper and lower levels, rather choosing specific levels for the vertical gradient. The reason is that we would not want our analysis to be level dependent. For

the barotropic component, note that one should not choose specific levels as by definition the barotropic component is the vertically integrated (through the entire troposphere) part of the flow (i.e., it is depth independent), as mentioned in the text. Note that not only eddy moisture fluxes show high correlation with changes in uyy, but ensembles accounting for ozone depletion show a significant increases in uyy relative to ensembles that lack ozone depletion. Such distinct behaviors are not presented in the metric for baroclinic instability.

Lines 179: After '. . . gradient of the mean relative vorticity' insert 'of the zonal flow'. (This connection has only been shown for uyy.)
By construction the zonal mean relative vorticity accounts only for the zonal flow (line 193).

We would like to emphasize again our gratitude to the reviewer who pointed us in important directions that have significantly improved this manuscript.

Sincerely,

Rei Chemke, Michael Previdi, Mark England and Lorenzo Polvani

**References**

Agosta, C., Amory, C., Kittel, C., Orsi, A., Favier, V., Gallée, H., van den Broeke, M. R., Lenaerts, J. T. M., Melchior van Wessem, J., van de Berg, W. J., and Fettweis, X. (2019). Estimation of the Antarctic surface mass balance using the regional climate model MAR (1979-2015) and identification of dominant processes. *The Cryosphere*, 13(1):281–296.

Cataldo, M., Evangelista, H., Simões, J. C., Godoi, R. H. M., Simmonds, I., Hollanda, M. H., Wainer, I., Aquino, F., and Van Grieken, R. (2013). Mineral dust variability in central West Antarctica associated with ozone depletion. *Atmos. Chem. Phys.*, 13(4):2165–2175.

Golledge, N. R. (2020). Long-term projections of sea-level rise from ice sheets. *Wiley Interdiscip. Rev.: Climate Change*, 11(2):e634.

Grieger, J., Leckebusch, G. C., Raible, C. C., Rudeva, I., and Simmonds, I. (2018). Subantarctic cyclones identified by 14 tracking methods, and their role for moisture transports into the continent. *Tellus*, 70(1):1454808.

Kay, J. E., Deser, C., Phillips, A., Mai, A., Hannay, C., Strand, G., Arblaster, J. M., Bates, S. C., Danabasoglu, G., Edwards, J., Holland, M., Kushner, P., Lamarque, J.-F., Lawrence, D., Lindsay, K., Middleton, A., Munoz, E., Neale, R., Oleson, K., Polvani, L., and Vertenstein, M. (2015). The Community Earth System Model (CESM) Large Ensemble Project: A Community Resource for Studying Climate Change in the Presence of Internal Climate Variability. *Bull. Am. Meteor. Soc.*, 96:1333–1349.

Levermann, A., Winkelmann, R., Albrecht, T., Goelzer, H., Golledge, N. R., Greve, R., Huybrechts, P., Jordan, J., Leguy, G., Martin, D., Morlighem, M., Pattyn, F., Pollard, D., Quiquet, A., Rodehacke, C., Seroussi, H., Sutter, J., Zhang, T., Van Breedam, J., Calov, R., DeConto, R., Dumas, C., Garbe, J., Hilmar Gudmundsson, G., Hoffman, M. J., Humbert, A., Kleiner, T., Lipscomb, W. H., Meinshausen, M., Ng, E., Nowicki, S. M. J., Perego, M., Price, S. F., Saito, F., Schlegel, N., Sun, S., and van de Wal, R. S. W. (2020). Projecting Antarctica's contribution to future sea level rise from basal ice shelf melt using linear response functions of 16 ice sheet models (LARMIP-2). *Earth System Dynamics*, 11(1):35–76.

Papritz, L., Pfahl, S., Rudeva, I., Simmonds, I., Sodemann, H., and Wernli, H. (2014). The Role of Extratropical Cyclones and Fronts for Southern Ocean Freshwater Fluxes. *J. Climate*, 27(16):6205–6224.

Rignot, E., Mouginot, J., Scheuchl, B., van den Broeke, M., van Wessem, M. J., and Morlighem, M. (2019). Four decades of Antarctic Ice Sheet mass balance from 1979-2017. *Proc. Natl. Acad. Sci. U.S.A.*, 116(4):1095–1103.

Simmonds, I., Keay, K., and Lim, E. (2003). Synoptic Activity in the Seas around Antarctica. *Mon. Weath. Rev.*, 131(2):272.